# Two Heads Are Better Than One: Exploiting Both Sequence and Graph Models in AMR-To-Text Generation

## Abstract

Abstract meaning representation (AMR) is a well-established special semantic representation language, which can capture the core meaning of a sentence with a syntax-irrelevant graph. AMR-to-text generation, which aims to generate a sentence according to a given AMR graph, is a well-studied task and has shown its helpfulness in various other NLP tasks. Existing AMR-to-text generation methods can be roughly divided into two categories, while either has its own advantages and disadvantages. The first one adopts a sequence-to-sequence model, especially a pretrained language model (PLM). It has good text generation ability but cannot cope with the structural information of AMR graphs well. The second category of method is based on graph neural networks (GNNs), whose advantages and disadvantages are exactly the opposite. To combine the strengths of the two kinds of models, in this paper, we propose a dual encoder-decoder model named DualGen, which integrates a specially designed GNN into a pre-trained sequence-to-sequence model. The GNN encoders are "pre-trained" by initializing with parameters from Transformer-based encoders in PLMs. We conduct extensive experiments as well as human evaluation and a case study, finding that it achieves the desired effect and yields state-of-the-art performance in the AMR-to-text generation task. We also demonstrate that it outperforms the most powerful general-purpose PLM GPT-4.

## 1 Introduction

Abstract meaning representation (AMR) is a well-established semantic representation language, representing sentence meanings as rooted, directed, and labeled graphs, free from syntactic idiosyncrasies (Banarescu et al., 2013). In an AMR graph, nodes depict entities, events, properties, and concepts, while edges denote relationships between nodes. Figure 1 exemplifies an AMR graph with two formats. AMR is valuable in NLP as it precisely captures text meaning without relying on specific language or syntax. This utility extends to cross-lingual tasks such as machine translation (Jones et al., 2012; Song et al., 2019), as sentences with the same meaning in different languages share identical AMRs. Furthermore, AMR's graph format simplifies problem-solving, allowing practitioners to manipulate input text's AMR graph directly (Liu et al., 2015; Huang et al., 2016).

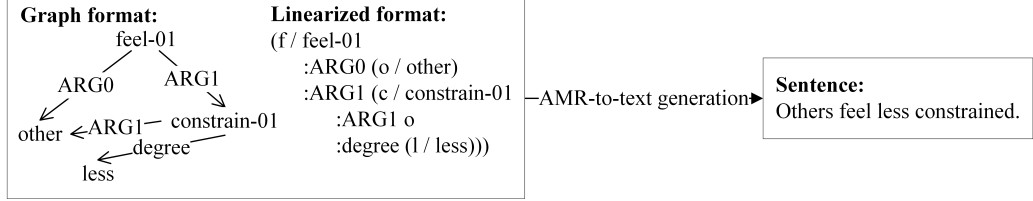

Figure 1: Illustration of two equivalent formats of an AMR graph and the AMR-to-text generation task. "ARG0", "ARG1", and "degree" are edge labels. In linearized format, nodes are denoted by abbreviations, e.g., "f" denotes "feel-01". The linearized format is indented for better readability.

AMR-to-text generation aims to generate a sentence with the same meaning from an AMR graph. It is a well-established task that is useful in various downstream applications, including text summarization (Liu et al., 2015; Takase et al., 2016), machine translation (Jones et al., 2012; Song et al., 2019), and information extraction (Zhang & Ji, 2021). Figure 1 illustrates AMR-to-text generation.

Previous studies of AMR-to-text generation employ two kinds of architectures. The first one is sequence-to-sequence (s2s) model, which uses a sequence encoder to process the linearized AMR graphs and a sequence decoder to generate text (Konstas et al., 2017; Cao & Clark, 2019). Benefiting from the strong generation ability of pretrained language models (PLMs) (Lewis et al., 2020; Raffel et al., 2020), recent s2s AMR-to-text models have achieved leading results (Ribeiro et al., 2021a; Bevilacqua et al., 2021; Bai et al., 2022). However, linearized AMR graphs used as the input of s2s models suffer from information loss compared with the original AMR graph, resulting in reduced performance of s2s models (Ribeiro et al., 2021b; Song et al., 2018; Beck et al., 2018).

The other kind is graph-to-sequence (g2s) models (Song et al., 2018; 2020; Beck et al., 2018; Guo et al., 2019) consisting of a graph neural network (GNN) encoder and a sequence decoder. Different from s2s models, g2s models can capture the full structural information of AMR graphs with its GNN encoder. Therefore, they usually outperform non-pretrained s2s models (Song et al., 2020), particularly for complex graphs. However, because g2s models cannot be pretrained on corpora, they exhibit weaker overall performance than pretrained s2s models and struggle to generalize to out-of-distribution data.

In this paper, to combine the strengths of both s2s and g2s models, we introduce DualGen, a dual encoder-decoder model for AMR-to-text generation based on PLMs. We use BART (Lewis et al., 2020) as the foundation model.[1] On the basis of the sequence encoder and decoder of BART, we add a GNN graph encoder to DualGen. In this way, DualGen is expected to absorb complete information of AMR graphs while benefiting from the strong text generation capabilities of PLMs.

Integrating a GNN encoder into a pretrained Transformer-based PLM is non-trivial. First, all existing AMR datasets are too small to adequately train a GNN encoder of a similar size as the sequence encoder from scratch. Second, no pretrained GNNs tailored for language tasks are available; prior studies employing both GNN and Transformer encoders for NLP tasks initiate GNN training from the ground up. To address these challenges, we design a specialized GNN encoder that can be initialized with PLM parameters. This encoder can seamlessly integrate with the PLM, sharing word embeddings with sequence encoders without adjusting vocabulary.

Experiment results on authoritative AMR datasets LDC2017T10 and LDC2020T02 demonstrate that DualGen outperforms the state-of-the-art method (Bai et al., 2022) and the most powerful PLM GPT-4 across multiple metrics. We conduct both quantitative and qualitative analyses, demonstrating that DualGen excels in processing graph structures while maintaining text generation quality on par with PLMs. We also find that DualGen particularly excels in handling complex graphs compared with s2s models, showing it possesses the advantage of both g2s models and s2s models. Moreover, we conduct a human evaluation and a case study that further validate these findings.

## 2    RELATED WORK

**AMR-to-text generation.** AMR-to-text generation entails transforming an AMR graph into its corresponding text. One approach for AMR-to-text generation employs a sequence-to-sequence (s2s) model, consisting of a sequence encoder and a sequence decoder. The initial neural model for this task (Konstas et al., 2017) uses stacked bidirectional LSTM, while recent studies adopt the Transformer architecture (Vaswani et al., 2017) and employ pretrained language models (PLMs). Ribeiro et al. (2021a) introduces adaptive pretraining, while Bevilacqua et al. (2021) explores linearization methods. Mager et al. (2020) incorporates a rescoring stage on top of the generation pipeline and explores joint probability. Bai et al. (2022) employs graph pretraining for BART. Inputting an AMR graph into the sequence encoder requires linearization. However, this linearization process leads to the loss of valuable graph structure information.

Another approach employs a graph-to-sequence (g2s) model, which is composed of a graph neural network (GNN) encoder and a sequence decoder. Various GNN encoders have been explored,

---

[1]We use BART as an example in this paper but DualGen is applicable to other Transformer-based PLMs.

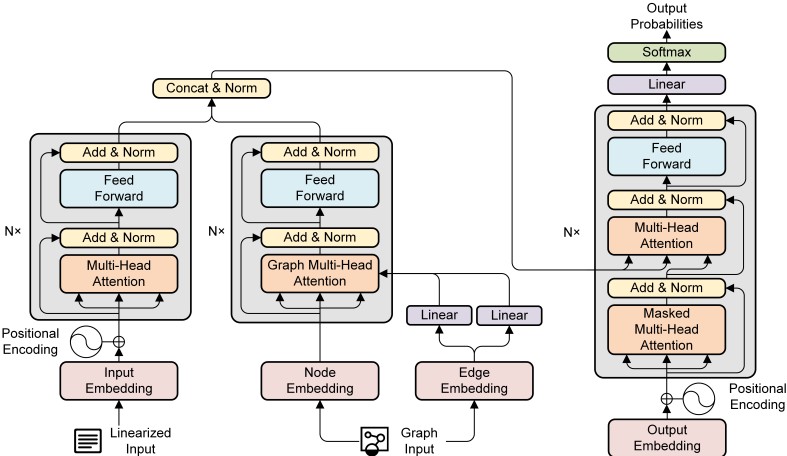

Figure 2: The architecture of the DualGen model.

including gated GNN (Beck et al., 2018), graph LSTM (Song et al., 2018), graph convolutional network (Guo et al., 2019), and graph attention network (Song et al., 2020; Koncel-Kedziorski et al., 2019; Cai & Lam, 2020). While the `g2s` architecture is well-suited for handling graphical input, it cannot process textual input. Consequently, it cannot be pretrained using textual data, which limits its natural language generation ability.

To combine the strengths of `s2s` and `g2s` models, Ribeiro et al. (2021b) employs a PLM-based approach, incorporating a graph convolutional network (GCN) adapter following the sequence encoder for better graphical data handling. Unlike DualGen which uses a dual encoder architecture, Ribeiro et al. (2021b) employs an un-pretrained GCN and only fine-tunes the GCN parameters while keeping others frozen. Later experimental results show the superiority of our method over this model.

**Dual encoder architecture.** The dual encoder architecture is widely used for various purposes. In generative models, prior work predominantly employs non-pretrained models. For instance, Junczys-Dowmunt et al. (2018) utilized two non-pretrained encoders and a decoder to recover machine translation errors. Zhang et al. (2021) applied two non-pretrained encoders and two non-pretrained decoders for abstract dialogue summarization.

In contrast, Dou et al. (2021) introduced a dual encoder-decoder model based on pretrained BART for text summarization. It comprises two Transformer encoders and a Transformer decoder, with each encoder independently processing full text and guidance signals as input. However, to the best of our knowledge, there has been no prior dual encoder-decoder model that simultaneously employs distinct architectures for the two encoders while utilizing pretrained models for both encoders.

For non-generative tasks, the dual encoder architecture is employed in tasks such as similarity measurement (Mueller & Thyagarajan, 2016; Yang et al., 2018), context-based candidate selection (Shyam et al., 2017), and information retrieval (Pang et al., 2017). In this setup, two input components are processed by distinct encoders and then assessed. To our knowledge, no prior research has employed the dual encoder architecture for AMR-to-text generation.

## 3 METHOD

In this section, we provide a detailed description of DualGen. We convert the AMR graph into a linearized and a graphical format (Section 3.1), which are then fed into our dual encoder-decoder model (Section 3.2). Following prior research, we employ a two-stage training (Section 3.3).

### 3.1 DATA PROCESSING

In preprocessing, we replace the nodes of an AMR graph with their original labels, omitting Prop-Bank (Palmer et al., 2005) indexes that indicate word semantic roles. For example, the node `f/feel-01` in Figrue 1 is transformed into `feel`.

Figure 3: Graph embedding.

To linearize, we use the DFS-based approach as per Bevilacqua et al. (2021). For tokenization, we use the original BART approach for both encoders, tokenizing each linearized AMR sequence and all nodes and edges in each AMR graph in the same way. This allows us to calculate sequence and graph embeddings, with shared parameters in the embedding module across the two encoders.

## 3.2 MODEL ARCHITECTURE

DualGen adopts a dual encoder-decoder architecture comprising a Transformer-based sequence encoder, a GNN-based graph encoder, and a Transformer-based sequence decoder, as depicted in 2. The sequence encoder and the graph encoder process the linearized AMR and the graph AMR, respectively.

**Sequence encoder:** The sequence encoder in DualGen is a Transformer encoder, initialized with BART parameters, as illustrated in the left gray box of Figure 2. It accepts the linearized AMR embedding with positional encoding as its input.

**Graph embedding:** The graph embedding comprises node and edge embedding, which share parameters with the sequence encoder embedding and the sequence decoder embedding. For any token $t$ in the vocabulary, its learned word embedding is denoted as $\mathbf{t} \in \mathbb{R}^{d_{\text{embed}}}$.

Given an AMR graph $\mathcal{G} = \langle \mathbb{V}, \mathbb{E} \rangle$, $\mathbb{V}$ is the node set, and $\mathbb{E}$ is the edge set. Each node and edge is labeled by a word or phrase, which is divided into multiple tokens during tokenization. These tokens are subsequently used to generate embeddings for nodes and edges. A node $v \in \mathbb{V}$ is denoted by $l_v$ tokens $t_1^v, t_2^v, \cdots, t_{l_v}^v$. An edge $e \in \mathbb{E}$ is denoted by $m_e$ tokens $t_1^e, t_2^e, \cdots, t_{m_e}^e$.

As Figure 3 shows, for any node $v \in \mathbb{V}$, its node embedding is the average embedding of all its corresponding tokens $\boldsymbol{v} = \frac{1}{l_v} \sum_{k=1}^{l_v} \mathbf{t}_k^v$

To facilitate two-way information exchange along edges, we introduce two linear projections from $\mathbb{R}^{d_{\text{embed}}}$ to $\mathbb{R}^{d_{\text{edge}}}$ for forward and backward edges, defined by parameter matrices $W^F, W^B$ and bias $\boldsymbol{b}^F, \boldsymbol{b}^B$. For an edge $e$ from node $s_e$ to $t_e$, its forward and backward edge embeddings are $\boldsymbol{e}^{fwd} = (\frac{1}{m_e} \sum_{k=1}^{m_e} \mathbf{t}_k^e)W^F + \boldsymbol{b}^F, \boldsymbol{e}^{bwd} = (\frac{1}{m_e} \sum_{k=1}^{m_e} \mathbf{t}_k^e)W^B + \boldsymbol{b}^B$

AMR graphs are acyclic, ensuring at most one edge connects any given pair of nodes. Therefore, the edge embedding is well-defined:

$$\forall s, t \in \mathbb{V}, \mathbf{e}_{s,t} = \begin{cases} \boldsymbol{e}^{fwd} & \text{if } s_e = s \text{ and } t_e = t \\ \boldsymbol{e}^{bwd} & \text{if } t_e = s \text{ and } s_e = t \\ \mathbf{0} & \text{otherwise} \end{cases} \quad (1)$$

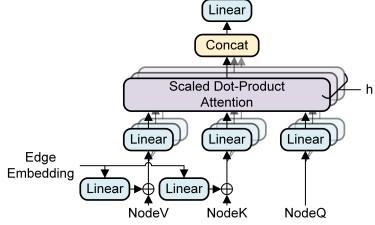

Figure 4: Graph multi-head attention.

**Graph encoder:** The graph encoder closely resembles the Transformer encoder, as shown in Figure 2. However, it incorporates a special multi-head attention mechanism for graphs, as depicted in Figure 4. The node embedding is $V^{\text{n}} = K^{\text{n}} = Q^{\text{n}} = \begin{bmatrix} \boldsymbol{v}_1 & \boldsymbol{v}_2 & \cdots & \boldsymbol{v}_{|\mathbb{V}|} \end{bmatrix}^{\top}$ and the edge embedding for a given node $v$ is $\mathbf{E}_v = \begin{bmatrix} \mathbf{e}_{v,1} & \mathbf{e}_{v,2} & \cdots & \mathbf{e}_{v,|\mathbb{V}|} \end{bmatrix}^{\top}$.

Inspired by the work of Song et al. (2020), we present a graph attention mechanism. To leverage edge information, we incorporate edge embeddings into the node value and node key components through two distinct linear projections from $\mathbb{R}^{d_{\text{edge}}}$ to $\mathbb{R}^{d_{\text{node}}}$ defined by parameter matrices $W_e^V, W_e^K$ and bias terms $\boldsymbol{b}^V, \boldsymbol{b}^K$., respectively. As discussed by Cai & Lam (2020), we treat the graph as a fully connected graph with specialized edge labels, which facilitates information exchange. The

formulation of this attention mechanism is as follows:

$$V_i = V^{\mathrm{n}} + \mathbf{E}_v W_e^V + \boldsymbol{b}^V, K_i = K^{\mathrm{n}} + \mathbf{E}_v W_e^K + \boldsymbol{b}^K, Q_i = Q_i^{\mathrm{n}} \tag{2}$$

$$\mathrm{GraphAttention}(Q, K, V)_i = \mathrm{Multihead\text{-}Attention}(Q_i, K_i, V_i) \tag{3}$$

The graph encoder is "pretrained" in a special way. It adopts a structure similar to the Transformer sequence encoder, allowing us to use pretrained BART parameters for initializing all of its components, except for the two additional linear projections. This initialization process is anticipated to enhance the language processing capabilities of the graph encoder.

**Hidden representation merging:** To merge the hidden representations from both encoders, we concatenate the two components and apply layer normalization (Ba et al., 2016).

**Sequence decoder:** The sequence decoder in DualGen follows the architecture of the Transformer decoder, as illustrated in Figure 2. It is initialized with pretrained BART parameters.

### 3.3  TWO-STAGE TRAINING

Considering the limited size of existing AMR datasets, which may not be adequate for training effective graph encoders, we employ a two-stage training strategy for AMR-to-text generation, aligning with prior research (Bai et al., 2022; Bevilacqua et al., 2021; Ribeiro et al., 2021a).

For the first stage, we employ model-generated silver data for pretraining purposes. Specifically, we randomly sample 200k entries from the Gigaword dataset (LDC2011T07) (Parker et al., 2011) and utilize the AMR parsing model `parse_xfm_bart_base` from amrlib (Jascob, 2020) to generate the corresponding AMR graphs. The model may generate some AMR graphs that are not accordant to AMR rules, and we remove these entries to ensure the correctness of the silver data set. For the second stage, we employ golden data from existing AMR datasets for fine-tuning.

## 4  EXPERIMENTS

We assess the performance of DualGen in comparison to state-of-the-art models on authoritative datasets. We investigate how graph complexity influences performance and evaluate the models' capacity to process graph structure through human evaluation. Additionally, we compared DualGen's performance with the most powerful PLMs including GPT-3.5 and GPT-4 (OpenAI, 2023).

### 4.1  DATASET

Following previous works (Bai et al., 2022; Ribeiro et al., 2021b; Bevilacqua et al., 2021) , we evaluate our model using the two most prevalent and authoritative AMR datasets, AMR2.0 (LDC2017T10)(Knight et al., 2017) and AMR3.0 (LDC2020T02) (Knight et al., 2020) datasets. Table 1 presents dataset statistics for both.

Table 1: Statistics of AMR2.0 and AMR3.0.

| Dataset | Train | Dev | Test |
|---------|-------|-----|------|
| AMR2.0  | 36,521 | 1,368 | 1,371 |
| AMR3.0  | 55,635 | 1,722 | 1,898 |

### 4.2  EVALUATION METRICS

Following previous works (Bai et al., 2022; Bevilacqua et al., 2021), we use three automated evaluation metrics: BLEU (Papineni et al., 2002), Meteor (Banerjee & Lavie, 2005), and chrF++ (Popović, 2015). We also perform human evaluation to assess language quality and semantic similarity.

### 4.3  COMPARED MODELS

We select representative methods for comparison: `g2s` models, `s2s` models which include the current state-of-the-art approach, and a hybrid model that merges both architectures. The compared models are (1) Guo et al. (2019), a `g2s` model that employs a densely connected graph convolutional network with an attention mechanism; (2) Song et al. (2020), a `g2s` model that utilizes a structure-aware Transformer encoder with vectorized edge information; (3) Ribeiro et al. (2021a), a `s2s` model based on PLMs [2]; (4) Bevilacqua et al. (2021), a `s2s` model based on PLMs that employs

---

[2]Ribeiro et al. (2021a) employs the original Bart model which shares the same architecture and training method as DualGen without graph encoders, with only minor vocabulary differences.

Table 2: Results of AMR-to-text generation for the AMR2.0 and AMR3.0 test sets. Models marked with † have g2s model architecture. We calculate results marked with ‡ as they are not reported in the original paper. The Silver Data column indicates the total number of data entries used for pretraining and finetuning. The best results within each dataset are denoted in bold.

| Dataset | Model | Silver Data | BLEU | Meteor | chrF++ |
|---------|-------|-------------|------|--------|--------|
| **AMR2.0** | Guo et al. (2019)† | 0 | 27.6 | 33.1$^{\ddagger}$ | 57.3 |
| | Song et al. (2020)† | 0 | 34.2 | 38.0 | 68.4$^{\ddagger}$ |
| | Ribeiro et al. (2021a) (Bart$_{large}$) | 0 | 43.5 | 42.9 | 73.9$^{\ddagger}$ |
| | Ribeiro et al. (2021a) (Bart$_{large}$) | 200k | 44.7 | 43.7 | - |
| | Bevilacqua et al. (2021) (Bart$_{large}$) | 200k | 45.9 | 41.8 | 74.2 |
| | Ribeiro et al. (2021b) (T5$_{base}$) | 0 | 44.0 | 41.9$^{\ddagger}$ | 71.2 |
| | Ribeiro et al. (2021b) (T5$_{large}$) | 0 | 46.6 | 42.8$^{\ddagger}$ | 72.9 |
| | Bai et al. (2022)(Bart$_{base}$) | 200k | 46.6 | 41.4 | 74.6 |
| | Bai et al. (2022)(Bart$_{large}$) | 200k | 49.8 | 42.6 | 76.2 |
| | DualGen (Bart$_{large}$) | 0 | 47.9 | 43.3 | 74.6 |
| | DualGen (Bart$_{large}$) | 200k | **51.6** | **44.9** | **77.0** |
| **AMR3.0** | Song et al. (2020)† | 0 | 37.9$^{\ddagger}$ | 39.4$^{\ddagger}$ | 70.8$^{\ddagger}$ |
| | Bevilacqua et al. (2021) (Bart$_{large}$) | 200k | 46.5 | 41.7 | 73.9 |
| | Ribeiro et al. (2021b) (T5$_{base}$) | 0 | 44.1 | 42.8$^{\ddagger}$ | 73.4 |
| | Ribeiro et al. (2021b) (T5$_{large}$) | 0 | 48.0 | 44.0$^{\ddagger}$ | 73.2 |
| | Bai et al. (2022)(Bart$_{base}$) | 200k | 45.9 | 40.8 | 73.8 |
| | Bai et al. (2022)(Bart$_{large}$) | 200k | 49.2 | 42.3 | 76.1 |
| | DualGen (Bart$_{large}$) | 0 | 49.5 | 43.9 | 75.7 |
| | DualGen (Bart$_{large}$) | 200k | **51.8** | **45.1** | **77.2** |

special linearization method and vocabulary; (5) Ribeiro et al. (2021b), a s2s model based on PLMs that includes a graph convolutional network adapter; (6) Bai et al. (2022), the state-of-the-art method, a s2s model based on PLMs that utilizes a unified graph pretraining framework.

## 4.4 SETTINGS

We use the BART-large model (Lewis et al., 2020) as the foundation model of DualGen. DualGen comprises 12 sequence encoder layers, 12 graph encoder layers, and 12 sequence decoder layers. The sequence encoder and decoder need minimal fine-tuning since they share BART's architecture; the graph encoder, with a different architecture, requires more fine-tuning. Consequently, we employ three distinct learning rates for the three components.

We select hyperparameters by validation set performance. For silver-data training, the model undergoes 6,000 steps over 20 epochs with updates every 8 steps, with a scale tolerance of 0.5 to filter out low-quality data. For fine-tuning, the model undergoes 13,000 steps over 65 epochs with updates every 4 steps. In both phases, the initial learning rates are $1 \times 10^{-6}$ for the sequence encoder, $4 \times 10^{-5}$ for the graph encoder, and $8 \times 10^{-6}$ for the sequence decoder. We use Adam (Kingma & Ba, 2015) as optimizer with $\beta_1 = 0.9, \beta_2 = 0.999$, and a clipping threshold of $0.1$.

## 4.5 MAIN RESULTS

Table 2 shows results on the test set of AMR2.0 and AMR3.0 under different models. The results on all three metrics demonstrate that our model DualGen outperforms all other methods. Compared to the state-of-the-art model (Bai et al., 2022), it achieves a 1.8-point improvement in BLEU, 2.3 points in Meteor, and 0.8 points in chrF++ on AMR2.0 dataset. Similarly, on AMR3.0, DualGen achieves a 2.6-point increase in BLEU, 2.8 points in Meteor, and 1.1 points in chrF++.

Models utilizing s2s pretrained language models consistently outperform non-pretrained g2s models. This suggests that pretraining on large corpora significantly enhances model performance, confirming the validity of our choice to employ PLM-based methods.

Table 3: Resuls of model failure analysis. All models are trained without silver data. # Failed indicates the number of failed cases. $\overline{\text{Edge}}$, $\overline{\text{Node}}$, $\overline{\text{Reentrance}}$, and $\overline{\text{Depth}}$ indicate the average number of edges, average number of nodes, average number of reentrance nodes, and average graph depth of the failed cases, respectively.

| Model | Architecture | # Failed | $\overline{\text{Edge}}$ | $\overline{\text{Node}}$ | $\overline{\text{Reentrance}}$ | $\overline{\text{Depth}}$ |
|---|---|---|---|---|---|---|
| Guo et al. (2019) | g2s | 751 | 19.37 | 18.55 | 1.82 | 3.39 |
| Ribeiro et al. (2021a) | s2s | 347 | 18.68 | 17.91 | 1.77 | 3.23 |
| DualGen | dual encoder | **260** | **18.22** | **17.65** | **1.57** | **3.10** |

Utilizing silver data, whether for pretraining or finetuning, consistently leads to better performance compared to methods that do not incorporate such augmentation. This highlights the effectiveness of our use of model-generated silver data.

Compared with Ribeiro et al. (2021a) which shares the same architecture and method as DualGen without graph encoders, DualGen consistently achieves superior performance. This underscores the effectiveness of incorporating a graph encoder in AMR-to-text generation. Further details of ablation studies can be found in Appendix A.

### 4.6 IMPACT OF GRAPH COMPLEXITY

To determine the robustness of DualGen across varying levels of graph complexity, and its effectiveness in processing graph structure information, we investigate how varying graph complexity influences the performance of g2s models, s2s models, and our model. For this investigation, we choose Guo et al. (2019) and Ribeiro et al. (2021a)[3] as the representative g2s and s2s models, respectively.

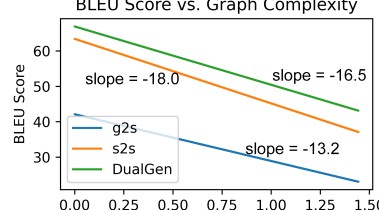

Figure 5: The impact of graph complexity on model performance.

A higher edge-to-node ratio suggests a more complex graph with intricate node relationships. To measure graph complexity, we use this ratio and conduct regression analysis to examine its connection with model performance, measured by the BLEU score. A steeper regression slope indicates improved data processing capacity, with a higher regression line indicating superior overall performance.

Figure 5 presents the regression outcomes. Among the three models, g2s excels with the highest slope for graphical data, while s2s struggles, showing the lowest slope. DualGen falls in between, performing well in processing graph structures, particularly when the edge-to-node ratio is high.

In terms of language skills, both s2s and DualGen perform similarly, surpassing g2s. Regression analysis confirms that the dual encoder-decoder architecture effectively enhances the model's ability to process graph structure while maintaining comparable language skills compared to PLMs.

### 4.7 MODEL FAILURES

To explore the shortcomings of the above three models Guo et al. (2019), Ribeiro et al. (2021a), and DualGen, we analyzed the failed cases. Entries with a BLEU score below 25 are considered failed.

The results are presented in Table 3. Compared with g2s and s2s models, in failed instances, DualGen exhibits fewer edges and nodes, reduced node reentrance, and shallower graph depth, indicating simpler graph structures. Given that the s2s model is the same as DualGen without graph encoders, the results imply that DualGen is less sensitive to intricate graph architectures. This underscores the efficacy of the graph encoder in processing AMR graphs.

### 4.8 HUMAN EVALUATION

To further assess the performance of the models, we conduct a human evaluation. Following previous work (Ribeiro et al., 2021b;a), we randomly select 100 AMR graphs from the AMR2.0 test set.

---

[3] We use the model introduced in Ribeiro et al. (2021a) without silver data pretraining, which is the original Bart model. It shares architecture and method with DualGen without graph encoder.

Table 4: Results of human evaluation on the AMR2.0 test set. Our model significantly outperforms comparison methods, as indicated by T-tests with a significance level of $p < 0.05$. The best language quality scores are underlined, while the best semantic similarity scores are in bold.

| Model | Architecture | Silver Data | quality | similarity |
|---|---|---|---|---|
| Song et al. (2020) | g2s | 0 | 8.22 | 8.01 |
| Ribeiro et al. (2021a) (Bart$_{large}$) | s2s | 0 | 9.26 | 8.26 |
| Bevilacqua et al. (2021)(Bart$_{large}$) | s2s | 200k | 9.11 | 8.35 |
| Bai et al. (2022) (Bart$_{large}$) | s2s | 200k | 9.42 | 8.57 |
| DualGen (Bart$_{large}$) | dual encoder | 0 | 9.29 | 8.59 |
| DualGen (Bart$_{large}$) | dual encoder | 200k | 9.38 | **8.98** |

Table 5: Results of large language models for AMR-to-text generation task for the AMR2.0 test set. The best results are highlighted in bold.

| Model | shot | BLEU | Meteor | chrF++ |
|---|---|---|---|---|
| GPT-3.5-turbo | 0 | 6.9 | 25.4 | 49.8 |
| GPT-3.5-turbo | 3 | 14.6 | 28.6 | 53.4 |
| GPT-3.5-turbo | 8 | 17.7 | 29.9 | 55.1 |
| GPT-3.5-turbo | 10 | 18.4 | 29.9 | 55.5 |
| GPT-3.5-turbo | 15 | 18.5 | 30.3 | 56.2 |
| GPT-4 | 15 | **30.8** | **36.7** | **64.7** |

Six annotators with an English background assessed these samples, providing scores on a scale of 0 to 10 for language quality and semantic similarity. Each entry was evaluated independently by three annotators to assess the performance of the six tested models. Further details of human evaluation settings and annotator agreements can be found in Appendix B.

Table 4 shows human evaluation results, where DualGen significantly outperforms comparison models.

For language quality, PLM-based s2s approaches consistently outperform the g2s method, indicating superior language proficiency. DualGen achieves language quality scores comparable to other PLM-based methods, affirming its similar language capabilities to PLMs.

Regarding semantic similarity, DualGen without silver data pretraining achieves a higher similarity score compared to other non-pretrained methods. DualGen with silver data pretraining significantly outperforms all other methods, demonstrating the clear benefits of the dual encoder architecture on AMR-to-text generation.

## 4.9 COMPARISON WITH THE MOST POWERFUL PLMS

Recent LLMs such as GPT-3.5 (OpenAI, 2021), GPT-4 (OpenAI, 2023), and LLaMA (Touvron et al., 2023) have demonstrated impressive language generation capabilities. We evaluate the performance of the most powerful LLMs GPT-3.5 and GPT-4[4] in AMR-to-text generation using the AMR2.0 test dataset. The results are presented in Table 5. Further details of the experiments can be found in Appendix C.

The results show that, although LLMs perform exceptionally well in many language-related tasks, they encounter difficulties in AMR-to-text generation without fine-tuning. We design prompts for in-context learning with a maximum of 15 shots due to the token limitation. GPT-4 with 15 shots outperforms all other LLM settings but lags significantly behind fine-tuned PLM methods. As a result, we conclude that LLMs are not proficient in AMR-to-text generation, with DualGen yielding significantly better results after training. Exploring smaller models for these specific tasks appears worthwhile as LLMs without fine-tuning cannot substitute these models.

---

[4]We use the ChatCompletion API https://platform.openai.com/docs/api-reference provided by OpenAI,

Table 6: Case study. The underlined text exhibits poor performance.

| AMR Graph | Text |
|---|---|
| (a / agitate-01
  :ARG0 (s2 / spring-up-02
   :ARG1 (s / scene
    :quant (m2 / many)
    :mod (h / heroic)
    :mod (t2 / tragic)
    :topic (a2 / and
     :op1 (s3 / spear
      :ARG1-of (s4 / shine-01))
     :op2 (h2 / horse
      :ARG1-of (a3 / armor-01)))
    :ARG2-of (s5 / stir-02))
    :location (m3 / mind
     :poss i))
  :ARG1 (s6 / string
   :poss (m / memory
   :poss (i / i))
   :mod (t4 / thing
    :ARG1-of (t3 / think-01)))
  :frequency (o / occasional)) | **Reference answer:** the thought-strings of my memory have been agitated from time to time - many heroic, stirring, and tragic scenes of shining spears and armored horses spring up in my mind. |
| | **Song et al. (2020):** occasionally, my memory has been touched by many heroic scene in my mind springing up in shiney spears and armored horses. |
| | **Ribeiro et al. (2021a):** my memory strings of thoughts are occasionally agitated by the stirring up of many heroic and tragic scenes of shining spears and armored horses in my mind. |
| | **Bai et al. (2022):** many heroic and tragic scenes that spring up in my mind of stirring spears and armored horses occasionally agitate the strings of thought in my memory. |
| | **DualGen:** occasionally, my memory's string of thoughts is agitated by the many stirring, heroic and tragic scenes of shining spears and armored horses that spring up in my mind. |

## 4.10 CASE STUDY

Table 6 presents a case study from the AMR2.0 test set, highlighting the superior performance of DualGen. It showcases sequences generated by both DualGen and the baseline g2s (Song et al., 2020) and s2s models (Ribeiro et al., 2021a; Bai et al., 2022), alongside the reference answer provided by the AMR2.0 dataset.

The answer generated by Song et al. (2020) contains grammatical errors, such as "many heroic scen" instead of "many heroic scenes". Furthermore, the phrase "in my mind springing up in shiny spears and armored horses" is unclear and ambiguous. These examples highlight the limited language proficiency of the g2s model.

The s2s PLM-based methods Ribeiro et al. (2021a); Bai et al. (2022) are proficient in generating grammatically correct and coherent sentences. However, Ribeiro et al. (2021a) overlooks specific entities, such as "spring up'. Both methods misinterpret edge relationships, failing to recognize that "heroic", "tragic", and "stirring up" should be juxtaposed. Furthermore, Bai et al. (2022) mistakenly employ "stirring" instead of "shining" to modify "spears".

Our model, DualGen, is free of grammatical errors, generates high-quality sentences, and accurately represents all node entities and edge relations. This demonstrates that our PLM-based model not only possesses strong language skills but also excels in managing graph structures simultaneously.

## 5 CONCLUSION

We explore a dual encoder-decoder architecture model for the AMR-to-text generation task. This model comprises a graph encoder, a sequence encoder, and a sequence decoder. Our model's architecture is specially designed to be compatible with Transformer encoder-decoder architecture, and all three primary components, including the graph encoder, can be initialized by PLMs such as BART (Lewis et al., 2020), GPT2 (Radford et al., 2019), and T5 (Raffel et al., 2020). We find that this dual encoder-decoder architecture enhances the model's capability to process graph structure information while maintaining language proficiency on par with PLMs. Our model surpasses the current state-of-the-art methods across multiple benchmarks for the AMR-to-text task.

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

## A  ABLATION STUDY

To further demonstrate the capabilities of each component within DualGen, we conducted an ablation study. This involved examining the performance of different model variations:

- DualGen w/o SE: DualGen without the sequence encoder;
- DualGen w/o GE: DualGen without the graph encoder;
- DualGen w/o GP: DualGen with the graph encoder trained from scratch, not initialized by Bart parameters.
- DualGen w/o SE w/o GP: DualGen without the sequence encoder, with the graph encoder trained from scratch, not initialized by Bart parameters.

We use GP to indicate graph pretraining, SE to indicate sequence encoders, and GE to indicate graph encoders. The outcomes for the above four model variants are presented in Table 7.

Table 7: Results of ablation study. We calculate results marked with $\ddagger$ as they are not reported in the original paper. The Silver Data column indicates the total number of data entries used for pretraining and finetuning. The best results within each dataset are denoted in bold.

| Dataset | Model | Silver Data | BLEU | Meteor | chrF++ |
|---------|-------|-------------|------|--------|--------|
| **AMR2.0** | DualGen w/o SE w/o GP | 0 | 0.0 | 1.0 | 3.4 |
| | DualGen w/o GP | 0 | 0.1 | 4.4 | 15.5 |
| | DualGen w/o SE | 0 | 22.1 | 31.4 | 58.7 |
| | DualGen w/o GE | 0 | 43.8 | 42.1 | 72.1 |
| | Ribeiro et al. (2021a) | 0 | 43.5 | 42.9 | 73.9$\ddagger$ |
| | DualGen | 0 | 47.9 | 43.3 | 74.6 |
| | DualGen | 200k | **51.6** | **44.9** | **77.0** |
| **AMR3.0** | DualGen w/o SE w/o GP | 0 | 0.0 | 1.3 | 3.3 |
| | DualGen w/o GP | 0 | 0.0 | 1.0 | 4.1 |
| | DualGen w/o SE | 0 | 22.2 | 31.6 | 58.2 |
| | DualGen w/o GE | 0 | 45.7 | 42.9 | 73.4 |
| | DualGen | 0 | 49.5 | 43.9 | 75.7 |
| | DualGen | 200k | **51.8** | **45.1** | **77.2** |

DualGen w/o SE w/o GP and DualGen w/o GP exhibit notably poor performance. This is because the AMR datasets are insufficient for training, given their limited size in comparison to the big size of the graph encoders. The training subsets of the AMR2.0 and AMR3.0 datasets comprise 36k and 56k entries, respectively. In contrast, the graph encoders contain 152M trainable parameters, akin in size to the Bart large encoders. In comparison, the full DualGen model encompasses 560M parameters, while the previously best-performing g2s model (Song et al., 2020) comprises a total of 62M parameters. Consequently, when fine-tuned on the AMR datasets, DualGen w/o SE w/o GP and DualGen w/o GP scarcely acquire meaningful information, consistently yielding a low BLEU score. This underscores the efficacy of our approach in "pretraining" the graph encoder in a specialized manner, initializing the GNN using Transformer encoder parameters.

DualGen w/o SE displays significantly lower performance compared to DualGen w/o GE and the full DualGenmodel. With only graph encoders, DualGen w/o SE encounters challenges in AMR-to-text generation. This is because the graph encoder is not intended to retain all information, particularly entity details of the nodes. Instead, it prioritizes structural information and facilitates information exchange between two nodes connected by an edge.

DualGen w/o GE performs similarly to the findings of Ribeiro et al. (2021a) without pretraining on silver data, aligning with our expected outcomes. Leveraging the strength of pretrained Transformer-based language models, the variant DualGen w/o GE notably outperforms the variant DualGen w/o SE.

The full DualGen model significantly surpasses DualGen w/o SE and DualGen w/o GE without individual encoders, highlighting the importance of incorporating both sequence and graph encoders for enhanced performance.

# B   HUMAN EVALUATION SETTINGS

For human evaluation, we use the test set of AMR2.0. We filter out sentences shorter than 30 characters to eliminate meaningless entries like "2004-10-09". Following this, we randomly pick 100 entries and assign them IDs from 1 to 100.

Six volunteer annotators, each with an English education background, carry out the annotation process. Three of them annotate entries 1 to 50, while the other three annotate entries 51 to 100.

Each entry $i$ contains a reference text $T_i$ from the AMR2.0 dataset and:

- the generated output $P_i^1$ of Song et al. (2020);

- the generated output $P_i^2$ of Ribeiro et al. (2021a);

- the generated output $P_i^3$ of Bevilacqua et al. (2021);

- the generated output $P_i^4$ of Bai et al. (2022);

- the generated output $P_i^5$ of DualGen without silver data pretraining;

- the generated output $P_i^6$ of DualGen with silver data pretraining.

For each assigned entry $i$, the annotator assigns scores $q_i^1, \cdots, q_i^6$ to rate the quality of sentence $P_i^1, \cdots, P_i^6$ and $s_i^1, \cdots, s_i^6$ to measure the similarity in meaning between $T_i$ and $P_i^1, \cdots, P_i^6$. The scores $q_i^1, \cdots, q_i^6, s_i^1, \cdots, s_i^6$ are integers ranging from 0 to 10 (inclusive). The rating criteria are outlined in Table 8.

Table 8: Rating criteria for human evaluation.

| Score | Criteria for Quality Score | Criteria for Similarity Score |
|---|---|---|
| 0 | The sentence has numerous grammar errors or contains many irrelevant words or phrases, making it completely incomprehensible to readers. | The information conveyed in the generated output text has no relevance to the information conveyed in the reference text. |
| 2 | The sentence has many errors in grammar, vocabulary, or word usage. Readers find it challenging to grasp the sentence's intended meaning. | The generated output mostly conveys information unrelated to the information in the reference text, only mentioning some of the concepts covered in the reference text. |
| 4 | The sentence has noticeable grammar, word, or phrase usage errors. Readers, through careful reading, can generally understand the main points of the sentence. | The generated output conveys some information that aligns with the reference text, but there are clear differences in their meanings. |
| 6 | The sentence has some grammatical errors or inappropriate word choices/phrases. The overall expression of ideas is somewhat coherent. Readers can generally understand the meaning. | The generated output mostly conveys the information covered in the reference text but either misses important details or includes some information not mentioned in the reference text. |
| 8 | The sentence contains a few grammar errors, uses words and phrases appropriately, expresses ideas coherently and naturally, and follows a logical structure that makes it easy for readers to understand the meaning. | The generated output conveys most of the information covered in the reference test but omits a few unimportant details or includes some unimportant information not mentioned in the standard text. |
| 10 | The sentence is free of grammar errors, uses appropriate words and phrases, expresses ideas coherently and naturally, follows a logical structure, and can be easily understood by readers in terms of its meaning. | The generated output conveys the same information as the reference text, neither omitting details nor including information not mentioned in the reference. |

## C  LARGE LANGUAGE MODELS EXPERIMENT SETTINGS

For the experiment on large language models, we use the OpenAI ChatCompletion API with the following settings:

We use the following system prompt to instruct the model:

Table 9: LLM settings.

| parameter | value |
|---|---|
| temperature | 0.01 |
| top p | 1.0 |
| n | 1 |
| frequency penalty | 0.0 |
| max tokens | 2048 |

```
System:
    Recover the text represented by the Abstract Meaning
    ↪  Representation graph (AMR graph) enclosed within
    ↪  triple quotes. Utilize only the information
    ↪  provided in the input. Output only the recovered
    ↪  text.
```

For few-shot prompting, we use the following format:

```
User:
    """
    (p / possible-01˜e.1
          :ARG1 (m / make-05˜e.2
                  :ARG0 (c / company :wiki "Hallmark_Cards"
                          :name (n / name :op1 "Hallmark"˜e.0))
                  :ARG1 (f / fortune˜e.4
                          :source˜e.6 (g / guy˜e.8
                              :mod (t / this˜e.7)))))
    """
Assistant:
    Hallmark could make a fortune off of this guy.
```

We evaluate GPT-3.5 using the entire AMR2.0 test set, whereas for GPT-4, we assess its performance by randomly selecting and testing 400 entries from the AMR2.0 test set.

