# OpenReview forum: "Two Heads Are Better Than One: Exploiting Both Sequence and Graph Models in AMR-To-Text Generation"
_ICLR.cc/2024/Conference — Submitted to ICLR 2024_

### Official Review · Reviewer_Koxu · 2023-10-17

**Soundness:** 3 good
**Presentation:** 2 fair
**Contribution:** 2 fair
**Rating:** 5
**Confidence:** 2

**Summary:**

The main contribution of the paper is to show that an ensemble of two established methods, sequence-to-sequence (s2s) and graph-to-sequence (g2s), performs better than either method by itself on two standard abstract meaning representation (AMR) benchmarks: AMR 2.0 and AMR 3.0.  The proposed ensemble method is called DualGen.

The paper shows results for a number of baselines including GPT-4.  It is interesting that the proposed method is considerably better than that on these benchmarks.

**Strengths:**

This paper is basically an incremental improvement over prior work (s2s and g2s).  It isn't surprising to see ensembles do well, but it is nice to see that that is the case.

The paper shows results for a number of baselines including GPT-4.  It is interesting that the proposed method is considerably better than that on these benchmarks.

**Weaknesses:**

This paper is written for an audience that is already familiar with the literature on AMR.  One would hope that the result (two heads are better than one) might generalize beyond this specific case, but there is little evidence or discussion of that.

I found the paper unnecessarily hard to read.  The lead sentence of the abstract doesn't make it clear that AMR is well-established in the literature, and many of the cited papers have hundreds/thousands of citations.

It may be useful to compare the description of the problem and the task in https://aclanthology.org/P18-1150.pdf (one of the key cited papers) with the description in this submission.  This submission does not make it as clear that AMR is an established concept in the literature.  Nor is it as clear what the task is.  The cited paper follows the standard formula where there is a section labeled "Experiments" with a subsection on materials (5.1 data).   The submission has a short section (4.1 Datasets) with references to LDC.  This made it clear that this is a standard problem in the literature. But by the time I get to the discussion of the dataset, it should be clear what the task is, and how much work there is on this task.

One would hope that the incremental improvement would be at least as good as the prior art in describing the problem and motivating it, but I found it easier to do that by reading the references than by reading this submission.

**Questions:**

Can you make this paper more accessible to an audience that is not already familiar with AMR?

The first part of the title (two heads are better than one) suggests that this result might generalize beyond this specific case (ensembling of s2s and g2s on two AMR benchmarks).  Is this paper limited to this specific case, or should it be of interest to a broader audience that may not be familiar with AMR?

---

> ### Author Response · Authors · 2023-11-21
> **Response to Reviewer Koxu**
>
> We thank the reviewer for the insightful feedback. Please find the answers to some specific questions below.
>
> > Can you make this paper more accessible to an audience that is not already familiar with AMR?
> >
> > I found the paper unnecessarily hard to read. The lead sentence of the abstract doesn't make it clear that AMR is well-established in the literature, and many of the cited papers have hundreds/thousands of citations.
> >
> > This submission does not make it as clear that AMR is an established concept in the literature. Nor is it as clear what the task is.
> >
> > The submission has a short section (4.1 Datasets) with references to LDC. This made it clear that this is a standard problem in the literature. But by the time I get to the discussion of the dataset, it should be clear what the task is, and how much work there is on this task.
>
> Thank you for pointing out that our paper is hard to read. In the first submitted version, we stated in the second sentence of the abstract that AMR-to-text is a well-studied task. In the revised version, we have added this point to the lead sentence.
>
> In the first paragraph, we further clarified the definition of AMR in the revised version, explicitly stating that this is an established concept. Figure 1 illustrates an example of AMR.
>
> In the second paragraph of the introduction, we introduced the AMR-to-text generation task more clearly in the revised version. We also use Figure 1 to illustrate an example of AMR-to-text generation, as previous studies have done. We then stated that our model DualGen is aimed to do AMR-to-text generation in the rest part of the Introduction more clearly.
>
> In Section 2 "Related Work", we introduced two types of previous methods on AMR-to-text generation, introducing that AMR-to-text is a widely-studied problem and there are a number of existing works exploring this topic.
>
> In response to your feedback, we have revised some expressions in the abstract and introduction to enhance clarity and better articulate the established task.
>
> > The first part of the title (two heads are better than one) suggests that this result might generalize beyond this specific case (ensembling of s2s and g2s on two AMR benchmarks). Is this paper limited to this specific case, or should it be of interest to a broader audience that may not be familiar with AMR?
>
> Dual encoder-decoder models have been widely used in NLP, as we stated in Section 2 "Related Works". Our main contribution is to:
>
> 1. Demonstrate that this architecture is also effective for AMR-to-text, since no previous work has employed a dual encoder-decoder model in this task.
> 2. Propose an innovative method for "pretraining" GNN encoders within the dual encoder-decoder framework, since previous studies showed that the pretraining technique is effective for language generative models and no prior work has explored pretraining a GNN for language tasks.
>
> AMR-to-text is a representative and valuable task for graph-to-text generation, and our method is useful in this setting. Because there is extensive work on dual encoder models consisting of un-pretrained GNN encoders, our method has the potential to be generalized beyond this specific case. We will try to add more experiments to prove that this method is also useful for other tasks later if our paper is accepted.

---

### Official Review · Reviewer_PF3i · 2023-10-29

**Soundness:** 3 good
**Presentation:** 3 good
**Contribution:** 2 fair
**Rating:** 6
**Confidence:** 4

**Summary:**

The paper proposes a new framework, DualGen, for AMR-to-text generation. The paper builds a new dual encoder-decoder model based on BART. Specifically, apart from the original BART sequence encoder, the paper introduces a new transformer-based graph encoder that takes in both node and edge embeddings as input. The graph attention mechanism incorporates edge embeddings into node representations. The paper then conducts experiments on both AMR 2.0 and AMR 3.0 datasets by comparing the proposed method with multiple state-of-the-art baselines. Following previous papers, the paper evaluates the results with BLEU, METEOR, and CHRF++. The paper also performs human evaluations. Additionally, the paper compared the proposed model with GOT-3.5 and GPT-4.

**Strengths:**

1. The paper proposes a new dual-encoder architecture, combining graph representation and linearized input. The paper also proposes a new graph attention mechanism incorporating edge embeddings into the node representations.
2. The paper did comprehensive experiments with automatic and human evaluations on two datasets. The proposed methods surpass other state-of-the-art models in both automatic and human evaluations. In addition, the paper analyzes the relationship between graph complexity and model performance. It shows that the proposed method can capture the complexities of the graphs more effectively. The proposed method outperforms GPT4. The paper includes a case study with detailed output for readers to compare.
3. The paper provides code, GPT4 results, silver data, etc.

**Weaknesses:**

1. The idea is a little bit incremental. The graph encoder is built upon Song et al., 2020. The idea of the dual encoder is also not new. For example, OntG-Bart (Sotudeh & Goharian, 2023) also uses a GNN and BART encoder-decoder framework to generate summarization. The dual encoder has also been applied to the multimodal domain (Li et al., 2021).
2. It would be better for authors to include an additional ablation study for the proposed method instead of only showing the final model. In this way, readers can have a better understanding of the contribution of each component. The analysis in section 4.5 is superficial and needs additional in-depth analysis. The authors can include additional quantitative analysis by analyzing the types of failures for each model.
3. Some parts of the paper are not clear. The paper fails to report the inter-annotator agreement for the human evaluations. The paper also includes typos. Typo in "Metor"-> METEOR


Li, J., Selvaraju, R., Gotmare, A., Joty, S., Xiong, C., & Hoi, S. C. H. (2021). Align before fuse: Vision and language representation learning with momentum distillation. Advances in neural information processing systems, 34, 9694-9705. https://proceedings.neurips.cc/paper/2021/file/505259756244493872b7709a8a01b536-Paper.pdf
Sotudeh, S., & Goharian, N. (2023, August). OntG-Bart: Ontology-Infused Clinical Abstractive Summarization. In Proceedings of the ACM Symposium on Document Engineering 2023 (pp. 1-4). https://dl.acm.org/doi/abs/10.1145/3573128.3609346

**Questions:**

Could the authors elaborate on how each encoder contributes to the overall performance of the model? Would the authors consider conducting an ablation study? Specifically, it would be insightful to demonstrate that the pretrained encoder effectively handles Out-Of-Domain (OOD) test cases, while the graph encoder adeptly captures structural information.

---

> ### Author Response · Authors · 2023-11-21
> **Response to Reviewer PF3i (1/3)**
>
> We thank the reviewer for the insightful feedback. Please find the answers to specific questions below.
>
> > The idea is a little bit incremental. The graph encoder is built upon Song et al., 2020. The idea of the dual encoder is also not new. For example, OntG-Bart (Sotudeh & Goharian, 2023) also uses a GNN and BART encoder-decoder framework to generate summarization. The dual encoder has also been applied to the multimodal domain (Li et al., 2021).
>
> We agree that dual encoders - one decoder is a common approach in NLP. However, we want to emphasize that our paper goes beyond the conventional use of a dual-encoder framework.
>
> In NLP, employing dual encoders with different architectures involves combining a pretrained Transformer-like sequence encoder with a non-pretrained encoder (e.g., GNN). This is also the case in OntG-Bart (Sotudeh & Goharian, 2023),  with a pretrained BART encoder and a non-pretrained GAT encoder.
>
> While GNNs are frequently employed in NLP dual encoder models, no prior work has focused on pretraining the GNN specifically for language-related tasks.
>
> In our work, **we introduce a distinctive approach by "pretraining" the GNN encoder**. This involves using pretrained parameters from the BART encoder to initialize our specially designed GNN, which is different from the established dual-encoder paradigm. This initialization is possible because we intentionally designed our GNN to resemble the Transformer architecture.
>
> Our work is significant because:
>
> 1. GNN is powerful in NLP tasks. However, there is no existing pretrained GNN for language tasks that provides a resource for direct use.
> 2. GNNs, unlike PLMs with Transformer architecture, are usually incompatible with being trained on a large corpus
> 3. As far as we know, no previous work has concentrated on pretraining the GNN component in dual-encoder models to enhance its effectiveness. This implies that our work has the potential to generalize to all Transformer-based dual encoder models in NLP where one of the encoders is a GNN.
>
> In response to your feedback, we have revised the Abstract, 1 Introduction, 3.2 Model Architecture, and 5 Conclusion in our paper to explicitly articulate our contribution to "pretraining" a GNN for language tasks.
>
> Reference:
>
> - Sotudeh, S., & Goharian, N. (2023, August). OntG-Bart: Ontology-Infused Clinical Abstractive Summarization. In Proceedings of the ACM Symposium on Document Engineering 2023 (pp. 1-4). https://dl.acm.org/doi/abs/10.1145/3573128.3609346
>
> > The idea is a little bit incremental. The graph encoder is built upon Song et al., 2020.
>
> **Although we used the idea of vectorized structural information in** **Song et al., 2020, there are essential differences between our encoder and Song's:**
>
> 1. We use a different method to generate node embeddings, utilizing Bart vocabulary and using the average value, rather than defining a special in-domain vocabulary for AMR
> 2. We use the Bart word embeddings and two additional linear projections for vectorized structural information, rather than using sequences of edges and specially defined tokens
> 3. We use a different attention mechanism, which resembles the original Transformer rather than Song et al., 2020.
>
> These differences ensure that the architecture of DualGen's graph encoder can be initialized with Bart parameters, and the graph encoder can integrate with the other part of the model seamlessly.
>
> Reference:
>
> - Song, Linfeng, et al. (2020). Structural Information Preserving for Graph-to-Text Generation. ACL. https://aclanthology.org/2020.acl-main.712/

---

> ### Author Response · Authors · 2023-11-21
> **Response to Reviewer PF3i (2/3)**
>
> > Could the authors elaborate on how each encoder contributes to the overall performance of the model? Would the authors consider conducting an ablation study? Specifically, it would be insightful to demonstrate that the pretrained encoder effectively handles Out-Of-Domain (OOD) test cases, while the graph encoder adeptly captures structural information.
> >
> > It would be better for authors to include an additional ablation study for the proposed method instead of only showing the final model. In this way, readers can have a better understanding of the contribution of each component.
>
> In our first version, we did perform an ablation study. We apologize for not explicitly mentioning that one baseline is equivalent to an ablation study.
>
> The ablation study involves three models besides DualGen: DualGen without graph encoders, DualGen without sequence encoders, and DualGen with un-pretrained graph encoders. We reported results for DualGen without graph encoders in the first version. We have included the outcomes for the other two variants in the revised version.
>
> 1. DualGen without graph encoders
>
> In Table 2, the baseline "Ribeiro et al. (2021a)" involves training BART with only vocabulary modifications, which is the same as DualGen without graph encoders. This model is also used in Section 4.6 where we analyze the impact of graph complexity. We also verified the results by training DualGen without graph encoders using our own code, producing similar results to Ribeiro et al. (2021a). Our omission in clearly stating that "Ribeiro et al. (2021a)" is the same as our model without graph encoders was an oversight on our part.
>
> 2. DualGen without sequence encoders
>
> The graph encoder in our model is designed not to take all information, but to focus on edge relations. The performance of DualGen without sequence encoders is worse (BLEU ~23) than the full model (BLEU ~48), aligning with our expectations.
>
> 3. DualGen with un-pretrained graph encoders
>
> The graph encoder of DualGen is large. The AMR train sets are small, which are inadequate to train the GNN from scratch. We have tested DualGen with un-pretrained graph encoders, which means the GNN is trained from scratch while the rest parts use Bart parameters. The result shows that the model learns nothing (BLEU < 2). This is not surprising considering the model size.
>
> In response to your feedback, we have modified sections 4.3 Compared Models and 4.5 Main Results to enhance clarity regarding Ribeiro et al. (2021a). Considering the limited information provided by the ablation study and the 9-page limit of ICLR, we included the ablation results, evaluated all by our own code rather than using results reported by Ribeiro et al. (2021a), in Appendix A in the revised pdf.
>
> References:
>
> - Ribeiro, Leonardo FR, et al. (2020) Investigating pre-trained language models for graph-to-text generation. NLP4ConvAI. https://aclanthology.org/2021.nlp4convai-1.20/

---

> ### Author Response · Authors · 2023-11-21
> **Response to Reviewer PF3i (3/3)**
>
> > The analysis in section 4.5 is superficial and needs additional in-depth analysis. The authors can include additional quantitative analysis by analyzing the types of failures for each model.
>
> We added Section 4.7 "Model Failure" to the revised pdf, examining the failures of three models: Guo et al. (2019) (g2s), Ribeiro et al. (2021a) (s2s), and our model. Ribeiro et al. (2021a) shares the same architecture and method as Bart, which is the same as our model without the graph encoders.
>
> We consider entries with a BLEU score lower than 25 as failed cases. We analyzed the graph size (indicated by edge number and node number), reentrance node number, and graph depth for failed cases of three models. We presented the results in Table 3 in the revised pdf and analyzed the results in Section 4.7 "Model Failure".
>
> References:
>
> - Guo, Zhijiang, et al. (2019). Densely connected graph convolutional networks for graph-to-sequence learning. TACL. https://aclanthology.org/Q19-1019/
>
> - Ribeiro, Leonardo FR, et al. (2020) Investigating pre-trained language models for graph-to-text generation. NLP4ConvAI. https://aclanthology.org/2021.nlp4convai-1.20/
>
> > Some parts of the paper are not clear. The paper fails to report the inter-annotator agreement for human evaluations. The paper also includes typos.
>
> Thank you for pointing out the need to add necessary details for the human evaluation experiment. In response to your feedback, we have included additional information on human evaluations in Appendix B. We also have corrected typos.

---

> ### Comment · Reviewer_PF3i · 2023-11-22
>
> Thank you for your response to my questions! The authors have addressed most of my questions. I raised my score to 6. However, the addition of a non-pretrained GNN variant ablation would greatly enhance the authors' claim of novelty.

---

> > ### Author Response · Authors · 2023-11-23
> > **Response to Reviewer PF3i**
> >
> > Thank you for your prompt response and raising the score! Please find the answers to specific questions below.
> >
> > > However, the addition of a non-pretrained GNN variant ablation would greatly enhance the authors' claim of novelty.
> >
> > We understand the importance of showcasing the novelty of our approach by including a non-pretrained GNN variant ablation. We've taken specific actions in response to your suggestions:
> >
> > To aid in clarity, we've established the following notation for discussions:
> >
> > - GP: Graph Pretraining
> > - SE: Sequence Encoders
> > - GE: Graph Encoders
> >
> > In our second submitted version, we mentioned in Appendix A that `DualGen w/o GP` performs extremely bad compared with the full DualGen model. We analyzed the reason, indicating that this is mainly because the AMR datasets are too small to sufficiently train the graph encoders from scratch. Therefore, graph pretraining is of great significance.
> >
> > In response to your feedback, we’ve added another variant `DualGen w/o SE w/o GP` in our third submitted version. To display the results more clearly, we’ve added the results of both `DualGen w/o SE w/o GP` and `DualGen w/o GP` to table 7 in the revised pdf. In the paragraph after table 7, we offered a detailed explanation of why these two variants performs extremely bad.
> >
> > For quick reference, the table below summarizes the performances on the AMR2.0 dataset. Conforming to our expectations, `DualGen w/o GP` performs badly, and `DualGen w/o SE w/o GP` performs even worse. The results indicate the usefulness of pretraining the graph encoders.
> >
> > | Model                 | Silver Data | BLEU     | Meteor   | chrF++   |
> > | --------------------- | ----------- | -------- | -------- | -------- |
> > | DualGen w/o SE w/o GP | 0           | 0.0      | 1.0      | 3.4      |
> > | DualGen w/o GP        | 0           | 0.1      | 4.4      | 15.5     |
> > | DualGen w/o SE        | 0           | 22.1     | 31.4     | 58.7     |
> > | DualGen w/o GE        | 0           | 43.8     | 42.1     | 72.1     |
> > | DualGen               | 0           | 47.9     | 43.3     | 74.6     |
> > | DualGen               | 200k        | **51.6** | **44.9** | **77.0** |

---

> > > ### Comment · Reviewer_PF3i · 2023-11-23
> > >
> > > Thank you for your clarification. I think all of my questions have been answered.

---

### Official Review · Reviewer_FACa · 2023-11-02

**Soundness:** 3 good
**Presentation:** 3 good
**Contribution:** 2 fair
**Rating:** 5
**Confidence:** 5

**Summary:**

In this paper, the authors focus on the AMR-to-text generation task.  To leverage the advantages of PLMs and GNNs, the paper proposes a dual encoder-decoder model called DualGen, which integrates a specially designed GNN into a pre-trained sequence-to-sequence model. The paper presents extensive experiments, human evaluations, and a case study, showing that DualGen achieves state-of-the-art performance in AMR-to-text generation tasks and outperforms the GPT-4.

**Strengths:**

1. The motivation for this paper is intuitive and the description of the proposed methodology is clear and comprehensible.

2. The proposed method exhibits compatible with any encoder-decoder architecture pre-train models.

3. The authors conduct an exhaustive comparsion, and the experimental results show the proposed method outperforms all previous work.

**Weaknesses:**

1. Actually, the multi-source structure (multiple encoders - one decoder) is a common approach in many NLP tasks. Even this paper does present some enhancements for the AMR-to-text generation task, the novelty of the proposed method appears to be somewhat constrained.

2. A commendable aspect of this paper is the authors' comparison of performance with GPT-4. However, considering that GPT-4 is a general model, the comparison may not be entirely equitable. It would be more appropriate to utilize an open-source large language model (e.g., Llama) for the experiment. This is crucial to verify the effectiveness of the proposed method in the context of Large Language Models.

**Questions:**

N/A

---

> ### Author Response · Authors · 2023-11-21
> **Response to Reviewer FACa**
>
> We thank the reviewer for the insightful feedback. Please find the answers to some specific questions below.
>
> > Actually, the multi-source structure (multiple encoders - one decoder) is a common approach in many NLP tasks. Even this paper does present some enhancements for the AMR-to-text generation task, the novelty of the proposed method appears to be somewhat constrained.
>
> We agree that multiple encoders - one decoder is a common approach in NLP. However, we want to emphasize that our paper goes beyond the conventional use of a multi-encoder framework.
>
> While GNNs are frequently employed in NLP multiple encoder models, no prior work has focused on pretraining the GNNs specifically for language-related tasks.
>
> In our work, **we introduce a distinctive approach by "pretraining" the GNN encoder.** We use parameters from BART to initialize the special GNNs in DualGen. This is different from the previous multi-encoder paradigm.
>
> Our work is significant because:
>
> 1. GNN is powerful in NLP tasks. However, there is no existing pretrained GNN for language tasks that provides a resource for direct use.
> 2. GNNs, unlike PLMs with Transformer architecture, are usually incompatible with being trained on a large corpus
> 3. As far as we know, no previous work has concentrated on pretraining the GNN component in multi-encoder models to enhance its effectiveness. This implies that our work has the potential to generalize to all Transformer-based multiple encoder models in NLP where one of the encoders is a GNN.
>
> In response to your feedback, we have revised the Abstract, 1 Introduction, 3.2 Model Architecture, and 5 Conclusion in our paper to explicitly articulate our contribution to "pretraining" a GNN for language tasks.
>
> > A commendable aspect of this paper is the authors' comparison of performance with GPT-4. However, considering that GPT-4 is a general model, the comparison may not be entirely equitable. It would be more appropriate to utilize an open-source large language model (e.g., Llama) for the experiment. This is crucial to verify the effectiveness of the proposed method in the context of Large Language Models.
>
> We believe GPT-4 outperforms all other unfine-tuned LLMs, so we didn't conduct experiments on other unfine-tuned models. As for fine-tuning open-source LLMs on AMR2text, we believe it's not a fair comparison considering the model size and training cost.
>
> Our reason for comparing DualGen with LLMs is that **using small models for AMR2text is still valuable** when LLMs are easily accessible, because un-finetuned LLMs perform badly, and fine-tuning LLMs is expensive. For better model performances, fine-tuning an LLM with a higher cost can always be an option.
>
> It's worth noting that the DualGen method can extend to LLMs as long as they are open-source and Transformer-based. We plan to add an experiment to fine-tune a dual-encoder Llama with LoRA if our paper is accepted. We apologize for not being able to complete this during this rebuttal session.
>
> In response to your feedback, we have revised the part “4.9 Comparison With The Most Powerful PLMs” in our paper to state our opinions more clearly.

---

### Author Response · Authors · 2023-11-21
**General Response**

We thank the reviewers for their insightful feedback. Following the reviewer's remarks, we made the following adjustments to the firstly submitted version:

| Goal                                                         | Adjustments                                                  |
| ------------------------------------------------------------ | ------------------------------------------------------------ |
| State our **contributions** more clearly, according to the suggestion from reviewer FACa and reviewer PF3i. | Add a sentence in the abstractAdd a few sentences in the second to last paragraph of Section 1 "Introduction"Add a few sentences in the last paragraph in the subsection "Graph encoder" in Section 3.2 "Model Architecture" |
| State our reason and conclusion for the **LLM experiments** more clearly, according to the suggestion from reviewer FACa. | Add a few sentences in the last paragraph of Section 4.9 "Comparison with the Most Powerful PLMs" |
| Explain **the relationship between Ribeiro et al. (2021a) and our model**, according to the suggestion from reviewer PF3i. | Add footnote 2 in Section 4.3 "Compared Models" and footnote 3 in Section 4.6 "Impact of Graph Complexity" |
| Add **ablation study**, according to the suggestion from reviewer PF3i. | Add Appendix A "Ablation Study"                              |
| Offer a **deeper analysis of our main results**, according to the suggestion from reviewer PF3i. | Add a paragraph to the end of Section 4.5 "Main Results"     |
| Analyze the **failed cases** of specific models, according to the suggestion from reviewer PF3i. | Add Section 4.7 "Model Failures"                             |
| Offer more details for the experiment settings and inter-annotator agreements for the **human evaluation**, according to the suggestion from reviewer PF3i. | Added Appendix B "Human Evaluation Settings"                 |
| **Clarify the task of this paper**, according to the suggestion from reviewer Koxu. | Modified some expressions in the Abstract and Section 1 "Introduction" |

---

### Meta-Review · Area_Chair_RMAF · 2023-12-10

**Metareview:**

In this paper, the authors focus on the AMR-to-text generation task. To leverage the advantages of PLMs and GNNs, the paper proposes a dual encoder-decoder model called DualGen, which integrates a specially designed GNN into a pre-trained sequence-to-sequence model.

All reviewers think the proposed method is incremental and the empirical results are not convincing.

**Justification For Why Not Higher Score:**

n/a

**Justification For Why Not Lower Score:**

n/a

---

### Decision · Program_Chairs · 2024-01-16

Reject